# Perspectives of Proteomics in Respiratory Allergic Diseases

**DOI:** 10.3390/ijms241612924

**Published:** 2023-08-18

**Authors:** Miguel Ángel Galván-Morales

**Affiliations:** Departamento de Atención a la Salud, CBS. Unidad Xochimilco, Universidad Autónoma Metropolitana, Calzada del Hueso 1100, Villa Quietud, Coyoacán, Ciudad de México 04960, Mexico; yumay0102@gmail.com; Tel.: +52-55-5483-7000

**Keywords:** proteomics, ELISA, mass spectrometry, liquid chromatography, Luminex, UniProt

## Abstract

Proteomics in respiratory allergic diseases has such a battery of techniques and programs that one would almost think there is nothing impossible to find, invent or mold. All the resources that we document here are involved in solving problems in allergic diseases, both diagnostic and prognostic treatment, and immunotherapy development. The main perspectives, according to this version, are in three strands and/or a lockout immunological system: (1) Blocking the diapedesis of the cells involved, (2) Modifications and blocking of paratopes and epitopes being understood by modifications to antibodies, antagonisms, or blocking them, and (3) Blocking FcεRI high-affinity receptors to prevent specific IgEs from sticking to mast cells and basophils. These tools and targets in the allergic landscape are, in our view, the prospects in the field. However, there are still many allergens to identify, including some homologies between allergens and cross-reactions, through the identification of structures and epitopes. The current vision of using proteomics for this purpose remains a constant; this is also true for the basis of diagnostic and controlled systems for immunotherapy. Ours is an open proposal to use this vision for treatment.

## 1. Introduction

Proteomics is the study of proteins in cells, tissues, and organisms during homeostasis or under specific conditions. The function of proteins in altered, natural, or specific conditions can be studied using proteomics. Mark Wilkins coined the term “proteome,” and proteomics is used particularly in medicine and molecular biology. These fields have made progress in identifying all known proteins and their post-translational modifications. Proteomics has also provided information that has been uploaded to databases and serves as algorithms in free programs today [1]. In the natural sciences, chemistry, medicine, and other fields of health research, proteomics has enabled the study of protein function, localization, production, and modification. It has also allowed a deeper understanding of the normal functioning of an organism, or, in contrast, the damage caused by its alteration [2,3].

The study of proteins in living systems, especially those directed at the expression or altered structures of proteins, aids in the search for protein markers, which can result in the design of diagnostic biomarkers and the discovery of new protein disease defects and effective follow-up therapies. Therapies can consist of the replacement of defective proteins, the production of epitope antagonists that trigger opposing responses, or the discovery of proteins that block receptors for cell signaling [4]. The study of protein anomalies through informatics program has also increased in frequency. The structure of proteins is predicted, allowing us to identify any potential alterations. At present, numerous strategies are used for the discovery, remodeling, and structural analysis of proteins for diagnostic, therapeutic, biochemical, molecular, and other purposes [5]. These strategies have also been used in conjunction with other techniques, such as protein engineering, pharmacogenomics, and other omics sciences. Since different disciplines are involved in proteomics, it is important to collaborate on shared public health problems and work towards a common goal [6].

For respiratory allergies and their molecular and clinical problems, technologies for their study, diagnosis, and treatment are required. Proteomics technologies provide the necessary tools to investigate the unknown molecules involved in the disease, the known proteins, their linkages and homologs, and other molecular determinants that produce hypersensitivity. Proteomics is one of the main approaches to studying the pathophysiology, alterations, pathways, and component elements of allergic disorders. In allergic diseases, the main objective of proteomics has been to discover new proteins, which affect the patient and are innocuous for others; the homology between them; their cross-reactions; the usefulness of immunotherapy; and their contraindications [7]. Multiple proteomics technologies have been applied in the field to describe the cellular behavior and the epitopes of both TCRs and IgEs. The most widely used tool to identify, quantify, and profile proteins is mass spectrometry (MS), although, in its early days, ELISA was the gold standard. In recent years, proteomics has made it possible to establish methodologies to describe various proteins from animals, plants, trees, and foods that cause respiratory allergies [8].

## 2. Global Characteristics of Allergies for Proteomics Studies

Proteomics is a field of science that is approached in an interdisciplinary manner and studies the complexity and dynamics of proteins in biological systems, mainly, as mentioned above, in structure and function. Proteomics is a powerful analytical and identification technology, but also has informatics program with advanced systems for modeling, predicting, and understanding the likely function of proteins within cells and organisms, airway microbiota, and alternative blocking therapies. Therefore, diseases that depend on protein expression or have protein mediators can be studied and analyzed in depth using proteomics [9].

The development of respiratory allergic diseases involves complex and multifactorial components. These components include intrinsic factors, such as genetic predisposition, sex, age, and ethnicity. Recent studies have identified approximately 120 genes associated with allergic diseases [10]. The proteomic approach is vital for understanding the immunological alterations in allergies, the protein characteristics of allergens, and the general characteristics of the host, as well as the characteristics of the antigen and the sequence of the epitope or epitopes of that protein. The main purpose of these approaches is to improve research tools, improve the status of patients, optimize diagnosis, make the therapies used more efficient, improve current therapies, and develop new therapies. Additionally, elucidating the epidemiological aspects, such as identifying who suffers from these allergic diseases and how they suffer, is important. By these means, we have identified which allergies are more frequent, and the age, sex, race, and population with the highest incidence [11].

Moreover, the second important aspect in the study of allergies is the extrinsic factors, which are factors attributed to the external environment, lifestyle, and comorbidities. Exposure to harmless allergens in the case of dust, cold, rodent urine, skin from mites, and other entities can cause the development of allergies. The constant exposure of key cells to allergens triggers or develops an exaggerated immune response. Moreover, exposure to tobacco smoke at any age, excessive hygiene, and low exposure to microbial pathogens, especially in childhood, leads to less maturation or display of lymphocytes and less stimulation, which suppresses the development, growth, and production of T-reg lymphocytes and increases the risk of developing an allergic response against unrecognized antigens that are safe in the general population [12].

The subtypes of Tregs in the immunosuppressive response are important for the control of allergic responses and valuable for the development of new therapies. Two main groups have been classified according to their origin: natural (nTregs) and induced (iTregs). The immune response to aeroallergens begins in the respiratory epithelium of allergy patients. In the beginning, molecules such as TSLP, IL-33, or IL-25 are secreted, and type-2 innate lymphoid cells (ILC2), which release interleukins IL-4, IL-5, IL-13, and IL-19, are activated. This occurs while other pro-inflammatory cytokines are secreted by the epithelium, among them MIP-1α and CCL19, which activate the macrophages and dendritic cells (DCs) that present the allergen [13]. Activated DCs internalize, process, and present the allergen to naive T lymphocytes, which proliferate to become Th2 cells, which secrete Th2 cytokines. From here, the recruitment of eosinophils, mast cells, and basophils to the target tissue is induced. In addition, Th2 lymphocytes present allergens to naive IgM + B lymphocytes, which differentiate into allergen-specific IgE + plasma cells. The IgEs bind to mast cells, basophils, and eosinophils that have already migrated to the tissue and will eventually release inflammatory mediators (see Figure 1) [14]. Therapies have already been proposed with human iTregs and potential markers, including GITR, ICOS, CD39, PD-1, and PD-L1, and it has been proposed that they could drive the development of Treg-based therapies against autoimmune, allergic, and other chronic inflammatory disorders [15].

The study of proteomics may provide new insights into the alleviation of the allergic response. Monoclonal antibodies directed to IL-5 or its receptor (IL-5R) have been created, and these antibodies have been incorporated into asthma treatment guidelines, as in the case of mepolizumab anti-IL-5 monoclonal antibody, benralizumab anti-eosinophilic humanized and afucosylated monoclonal antibody (IgG1, kappa), and others where modifications have been carried out with proteomics (see Figure 1) [15]. Moreover, other treatment alternatives include immunotherapy with monomers of the causative protein, blockades of the type-1 hypersensitivity pathway, and the modification of the iTreg response with the production of IgGs instead of IgE [16]. 

Some of the main topics associated with allergic diseases that are currently being investigated with proteomic methods are: (1) the antigens that trigger the response, (2) the epitopes of the immunoglobulins that recognize them, and (3) cell migration receptors. These proteins are studied because they are responsible for the onset of hypersensitivity; without these three processes, there would be no allergic response. Although there are many other methods for diagnosis and treatment, markers that identify what type of allergy it is (hypersensitivity, pseudo allergy, or allergies) and what protein component is the cause of the allergy in the patient are lacking. 

In this case, there are important advances with proteomic methods where de novo proteins have been described in some pollens (for example, Fra e1 and Syr v 1 from Fraxinum). Protein antigens that trigger responses with their epitopes, antibodies with their paratopes, and signaling pathways in the type-2 inflammatory response are currently being investigated. Due to their importance in the phenomenon, mutated proteins or only the portion that is recognized are given as therapy and have diagnostic marker potential (see Figure 1 (4)). Proteomics, which is based on several techniques, is probably the most informative tool that can be used to resolve the challenges associated with allergic respiratory diseases.

## 3. Proteomic Technologies for the Study of Respiratory Allergies

There are approximately one million protein post-translational modifications, although it has been calculated that the human genome codes for approximately 30,000 proteins. Under environmental conditions, protein expression is modified and sometimes altered. However, epitopes are regularly conserved and an individual can have an allergic reaction depending on the context, and occasionally the immune response is increased by adjuvants [17]. In proteomics, there are multiple techniques for analyzing proteins and studying the quantitative and qualitative expression of proteins. The protein involved in the disease and its modifications can be studied in allergic patients, and the differences can be established between patients and healthy individuals, which guides us to know what is wrong in the individual. New proteins, membrane proteins, proteins in signal transduction pathways, and phosphorylated proteins can be identified in blood or in solution, and the structure of proteins can be identified by Rx crystallography or Protein Data Bank informatics program predictions [18].

Proteomics studies proteins in their different forms of expression, and they are usually detected using antibodies, but there are also techniques that separate and measure the size and composition (amino acids) of charged ions, such as chromatography and mass spectrometry. In addition to gel electrophoresis, one-dimensional (1D) and two-dimensional (2D) or 2D differential gel electrophoresis (2D-DIGE) and protein microarrays may or may not use antibodies [19]. High-throughput sampling without gel is also performed, such as when using multidimensional technology for proteins where specific recognition antibodies are used, such as ELISA and Luminex. The latter uses antibodies for the detection of proteins. Mass spectrometry may have some variants—for example, using atomic absorption, isotope labeling of affinity, or isobaric labeling for relative and absolute quantification, or in tandem and multidimensional protein identification technology (MudPIT) that combines it with liquid chromatography [20]. Additionally, in this combination, shotgun/proteomic analysis is used, which offers an indirect measurement of proteins after the proteolytic digestion of intact proteins and is subjected to LC–MSMS/MS analysis [21].

### 3.1. ELISA

The enzyme-linked immunosorbent assay is a technique that detects and measures the amount of a substance in a solution. Antibodies are used against particular proteins, which in this case are the epitopes of the allergens. There are direct, indirect, inhibitor, competitive, sandwich, and ELISPot ELISAs [22]. This technique has been used for allergy diagnosis, and type-1 hypersensitivity reaction is one of the main mechanisms in the development of respiratory allergy. This IgE-mediated mechanism involves a sensitization step with the arrival of the allergen, presented by APC to naïve T cells that will transform into Th2 cells that produce IL-4, IL-5, and IL-13 cytokines. They will also activate B cells and induce the production of specific IgE, which binds to FcεRI (this receptor is a member of the Ig superfamily), an antibody isotype involved in signal transduction on mast cells and basophils. This triggers a complex signaling cascade, leading to the release of inflammatory and vasoactive mediators [23]. Any of the steps of the reaction can be identified by ELISA; in fact, in the past and to date, identification kits and microarray chips have been based on diamond carbon solid-supported ELISA for the high-density immobilization of antigens (ISAC) [24,25] Abnova (Heidelberg, Germany); ImmunoCap (Phadia, ThermoScientific, Sweden) for the quantification of total and specific IgE; Immulite (Siemens, Munich, Germany); and Hytec-288 (Hycor, Garden Grove, CA, USA). In addition, allergen-specific IgE levels have been compared between the Immulite 2000 and ImmunoCAP systems for the identification of six inhaled allergens [26]. Another form of allergen diagnosis without IgE quantification, which has recently been reported, is based on the quantification of allergen-specific Th2 cells in blood serum and the quantification of IL-5, and it works against inhalant allergens [27].

Aeroallergens are airborne antigens that enter through the respiratory tract, even though they can also enter the conjunctiva and skin. This provokes the production of specific IgE antibodies, which is why studies have been directed to the IgE produced by the individual. By confronting the individual with the allergen, it is possible to determine what he/she is allergic to and thus make the diagnosis. The most important aeroallergens due to their impact are pollens, mites, fungi, the hair and urine of dogs, cats, and rodents, cockroaches, and molds [28,29,30,31].

ELISPot is used to investigate hypersensitivity reactions to drugs, and the confirmation of causality frequently facilitates the decision on whether to continue therapy. This technique is performed during monotherapy because the identification of the drug causing the reaction is often difficult, but this method can be used for drug challenge tests. Therefore, laboratory tests are of great interest since they can elucidate the causal diagnosis without putting the patient at risk [32]. Other tests of this technique have been used to identify immunoglobulins and hypersensitivity reaction cytokines and to track the cells of this response. An example of the usefulness of this technique was found when identifying the persistence of memory Th0 cells after activation by exposure to Japanese cedar (Cryptometria japonica) and Japanese cypress (Chamaecyparis btuse) pollens. After allergen exposure, Th0 cells were converted to Th2 cells, which were eventually thought to disappear until subsequent exposure. Two recombinant hybrid peptides were used for the ELISPOT assay: one with seven CD4 T cell determinants (from Cry j 1 and Cry j 2 and others from Japanese cedar) and fourteen other peptide CD4 T cell determinants (from Der f 1 and Der f 2 for mite allergies). These peptides were restricted to class II and recognized only Th cells. In the assay, magnetic beads with conjugated antibodies were used to detect responding cells. When CD4 cells were removed, ELISPOT spots disappeared; however, after CD8 depletion, the number of spots was equal to that of whole count of peripheral blood mononuclear cells. CD28 depletion also caused the spots to disappear. These results demonstrate that the present ELISPOT assay using a hybrid peptide was restricted to CD4. This proved that Th2 cells respond to specific allergens. The tests were performed on patients exposed for multiple periods of 8 months, and after 5 months without exposure, they had traces of cells (memory Th2) [33].

### 3.2. Luminex

Luminex is another technique that is similar to ELISA in that it broadly detects proteins. The assay is performed on well plates containing color-coded beads and specific antibodies that capture the target analyte. Biotinylated antibodies are then added to the sample that binds to the analyte, forming an antibody–antigen–antibody sandwich through the streptavidin signal conjugated with phycoerythrin (PE). The samples are entered into the equipment, and the beads will be detected based on a dual laser flow (the analyzers are Luminex 200 or FlexMap) [34].

An important contribution of the technique and a concern expressed in this article is that the full identification of allergenic IgE epitopes is essential for the development of new methods for allergy diagnosis, treatment, and prognosis. Luminex quantifies and validates a bead-based epitope assay (BBEA) that, by multiplexing epitopes and processing multiple samples, allows large experiments to be performed in a short time, using minimal amounts of patient blood. The epitope-targeted identification of IgE is a sensitive biomarker of allergy and can also be used to predict the severity and phenotypes of allergy, as well as a quantification of the relationship between epitope-specific IgE and IgG4 quantification, to improve our understanding of the immune mechanisms underlying allergic sensitization [35].

The duality consists of one laser classifying the bead and the other classifying the analyte to which it is bound. There are lasers that act with greater clarity depending on the equipment. This technique is most useful if the different proteins in pollen that can activate the hypersensitivity response are known, so that we are able to identify the several epitopes in the pollen protein that trigger a response or the cytokines involved in allergy.

### 3.3. Western Blotting

The most commonly used techniques for the separation and isolation of proteins is Western blotting, and it can identify proteins by molecular weight and specific antibodies. In addition, proteins that have been modified can be studied with this method through the development of a specific antibody for that modification. There are antibodies that only recognize certain proteins when they are phosphorylated in tyrosine, and this method is widely used in Western blotting. SDS–PAGE is a method that involves the use of gel electrophoresis to separate the sample proteins and then transfer them to a membrane and locate the protein by its weight, or it uses the aforementioned antibodies to tag the protein and quantify it [36]. Additionally, immunoprecipitation is a variant of the technique used to extract proteins from ligands. The immunoprecipitation of protein complexes (Co-IP) is a technique used to locate a specific protein without an antibody for that protein. The localization of phosphorylated proteins is commonly used to locate proteins in intracellular signaling pathways since the protein activation of the cascade serves as a signal to activate transcription factors [37].

Many pollen allergenic proteins were identified by this method in the 1960s. The examination of an allergen called II-A and II-B (from lolium perenne) by starch gel electrophoresis in borate-6 m. urea at pH 8.5 revealed the presence of several discrete components, confirming the heterogeneity suggested by N-terminal amino acid analysis. Thus, SDS-PAGE procedures were no longer as rudimentary [38].

Also, the discovery of IgE was also in the 1960s, using sera from patients atopic to ragweed pollen. This was fractionated by chromatography on DEAE column cellulose and underwent gel filtration, sucrose density gradient ultracentrifugation, and agarose gel electrophoresis [39].

Since then, methods have been based initially on radioimmunoassays, and later on, enzyme-immunoassays and Western blotting were developed, capable of detecting these antibodies in the serum of allergic patients and the pollen proteins causing the reaction. It has allowed, together with mass spectrometry (MS), for the discovery of allergens in allergenic sources with cross-reactivity, which was not thought to be possible. An example is the discovery of a class III chitinase in coffee (Cof a 1) [40]—a glutamic acid-rich protein (Man e 5) that cross-reacts with Hev b 5 from latex [41].

This method can be used to identify the epitopes that bind to IgEs. It is also a tool that uses high-performance liquid chromatography (HPLc) and Western blotting with specific antibodies to detect epitopes and de novo proteins in pollens, urine, and the environment. Of the two-dimensional methods, 1-DE and 2-DE, the first to isolate proteins by 1-DE based on their molecular mass can be used to verify the purity of samples, check the purification of proteins, and calculate the unknown molecular weights, whether in the case of pollen proteins not yet described or of foods that cause allergies and the causative proteins are unknown [42]. The difference in 2-DE is the separation of the protein; it is based on the molecular mass and the isoelectric point, identifying different forms of proteins [43].

On the other hand, allergies to aeroallergens from tree, weed, or grass pollens have been best characterized by liquid chromatography, mass spectrometry, sequencing, and Western blotting. For example, the characterization of the panallergens profilin and polcalcin from ash pollen has shown that the amino acid sequences of profilin and polcalcin from ash pollen have a high degree of homology from different sources, with a structural similarity to those obtained from other members of these families. It also showed that the recombinant allergens were equivalent to those of the natural pollen counterparts of Fra e 2 and Fra e 3 (F. excelsior) and could be used in clinical diagnosis [44]. The families of these panallergen proteins are named Porphyrins, Procalcin, Vicillins, PR10, PR14, and LTP, and others are labelled as the super family similar to various allergenic proteins. This description through families has also allowed for the identification of different proteins from a single allergenic source, homologous proteins in different species (animal or plant), and different epitopes of an allergenic molecule [45,46,47,48].

### 3.4. Liquid Chromatography

This method allows for the isolation and purification of proteins in most samples in a production range of a few nanograms to picograms. Obtaining large amounts of protein from a sample produces more reliability and accuracy in the results. There are size-exclusion chromatography (SEC) or hydrodynamic techniques that consist of a gel compound for the retention of proteins by filtration and by solvents. Additionally, ion exchange chromatography (IEC) allows for the separation of the molecule by the nature of the charge [49]. High-performance liquid chromatography (HPLc) is the only separation method that satisfies the standards for obtaining samples without the excessive degradation of the proteome or sample, assisting with the rapid purification and isolation of peptides. Affinity chromatography is the protein-separation process employed by various techniques and works according to the immobilized ligand [50]. One method that does not use antibodies is mass spectrometry. It is an analytical technique that allows for the study of various compounds in nature, as well as for obtaining qualitative or quantitative information. In this way, it is possible to obtain information on the molecular mass of the analyzed compound and its structure.

The initial identification of animal allergens in the 1970s was highlighted by liquid chromatography and Western blotting [51]. Dogs and cats are sources of allergens through their fur, saliva, dander, serum, and urine. The detection of sensitization to cats has been easier than the detection of sensitization to dogs. The purification and characterization of the dog component protein Can f 1 occurred in 1973 in a study conducted by Ohmanjr, J. et al.—three separate fractions turned out to be the allergens, which corresponded to Can f 1. Subsequently, seven allergens have been reported for dogs (Can f) and eight for cats (Fel d) [51,52,53]. Furthermore, the purification and partial characterization of the main allergens from guinea pigs and rabbits were conducted, resulting in the identification of Cav p 1, Cav p 2 and Cav p 3 [54,55]. Moreover, Rat n 1 from rats [56] and allergens have been reported, and Mus m 1 and Mus m 2 from mice were identified by liquid chromatography [57,58]. Others that have also been identified are for dust mites (Der p1, Der f1), for cats (Fel d 1), for dogs (Can f 1), for cockroaches (Bla g 1 and 2) [59], and for mold (*Cladosporium*, *Penicillium* and *Aspergillus*) [60]. Almost all of these respiratory allergens have been characterized by dynamic markers using Western blotting, HPLc, and polymorphism, and have been identified by ELISA. Here, in addition we used IEDB Analysis Resource (Discotope 2.0) to predict the conformational epitopes of the heterogeneous molecules resulting from Mus m1, in order to standardize the murine lipocalin family [61].

### 3.5. Mass Spectrometry for Tree Allergen Homology

Allergic diseases caused by pollen represent the most frequent type-1 allergies and affect up to 30% of the population in industrialized cities. The trees that most frequently produce pollen allergy are: phagales (birch, alder, beech, oak), lamiales (ash, thunder, olive, lilac), pinales (Cypress, Japanese Cedar, Juniper), and proteales (Plantain and Ficus). For the description of the characteristics of these allergens (see Table 1), the most commonly used tools are Western blotting, mass spectrometry, and sequencing [62].

This analysis method quickly elucidates the protein sequence and its weight. MS is mainly based on ionizing the protein, so the spectrometers have an ion source, a detector, and an analyzer. To ionize a molecule, there are several methods, since liquid chromatography separates the molecules and spectrometry analyzes it by gases or is combined with liquid chromatography, as in the case of electrospray ionization (ESI).

In contrast, there is matrix-assisted laser desorption ionization (MALDI), which employs a matrix to which the sample adheres or absorbs and, under certain conditions, is localized by the laser beam. There is also another simpler call: a time-of-flight analyzer (TOF) based on the capture of ions traveling a distance, thus increasing resolution and improving the separation of the beams, which accurately determines the mass without the need for a magnetic or electric field [63]. This method can be combined with MALDI/TOF and others, such as the LC–MSMS/MS method, by high-resolution MRM. The latter helps us to find different molecules with the same mass, as well as identify many of the antibodies of polyclonal origin that develop in the organism. However, this technique seems to be too sophisticated for protein molecules, because there are simpler methods with the same specificity. Currently, they are used in the study and differentiation of lipids [64].
ijms-24-12924-t001_Table 1Table 1Pollen allergens.TreesAllergensRef.**Birch**Bet v 1, Bet v 2, Bet v 3, Bet v 4, Bet v 5, Bet v 6[62,63,65]**Alder**Aln g 1, Aln g 2, Aln g 3, Aln g 4[65]**Beech**Fag s 1[66]**Oak**Que a 1[67]**Ash**Fra e 1. Fra e 2, Fra e 3, Fra e 6, Fra e 10, Fra e 11, and Fra e 12[44,68]**Thunder**Lig v 1[69,70]**Olive**Ole e 1 to Ole e 12 [71,72]**Lilac**Syr v 1, Syr v 2, Syr v 3[62,73]**Cypress**Cha o 1[74,75]**Japanese cedar**Cry j 1, Cry j 2[2,76]**Juniper**Jun a 1, Jun a 2[2,77]**Plantain**Pla a 1 to Pla a 4[78]**Ficus**Fic c 1[72]**Walnut**Jug r 5[79]

The characterization of allergenic proteins from different pollens has been ongoing for several decades. Most studies have focused on identification, homology between families, and allergenic sequences, as there is cross-reactivity between allergenic proteins due to their similarity. Most studies aiming to reveal homology have used techniques called ELISA kit, Western blotting, liquid chromatography, and mass spectrometry. There are proteomics studies with similar approach patterns for almost all allergens.

There are a lot of data on homologies, and one of the best examples of a large dataset is that of Barbara Casos et al., who investigated 66 patients with rhinoconjunctivitis and/or asthma who had positive skin tests and/or specific IgE determination to olive and grass pollen. However, only those patients positive for the IgE of minor allergens underwent cross-reactivity and proteomic analysis, which revealed the presence of 42 common proteins in grasses and olive pollens. However, sensitization to olive and grass pollen is not due to cross-reactivity [72].

Allergens homologous to Bet v 1 (a major allergen of alder and hazel pollen) constitute a group of defense proteins (PR-10) for plant allergies associated with birch pollen and cross-reactivity with alder and hazel. The purification and characterization of the homology and cross-reactivity of Bet v 1 to three of the best-known groups of profilins and lipid transfer proteins (LTPs) were carried out by hybridization-cross-linking and SDS-PAGE [79].

It has been found that the allergic protein of cypress Cha o 1 is glycosylated and that the amino acid sequence is highly homologous with that of Japanese cedar pollen allergen (Cry j 1)—this was proven using proteomics tests. The high-amino-acid sequence identity and shared protein functions within each allergen group described by SDS-PAGE are described in detail for *Cryptmeria japonica* Japanese cedar Cry j 1-Cry j 3; Cry j cellulase; Cry j 7 vs. *Chamaecypari obtusesa* Japanese cypress Cha o 1-Cha o 3; and other allergen species. This means there is a high homology with cross-reactivity present during pollinosis in the presence of any of the species [75]. It has been further found that Cha o 1 is glycosylated and the amino acid sequence is highly homologous with that of Japanese cedar pollen allergen (Cry j 1); however, MS/MS analysis confirms the homology of cedar pollen (Jun a 1 and Cry j 1) and Arizon cypress, which with the N-glycan of Cha o1 is GlcNAc2Man3Xyl1-Fuc1GlcNAc2 [80].

Homology between proteins causes cross-reactivity, and detecting cross-reactivity with other non-protein structures has not been easy; however, the presence of glycosylated forms has been the subject of research. An example is a study by Maria Torres et al. in which they purified and identified Fra e 9 as a new allergen (β-1,3-glucanase) from ash pollen and the olease family, finding that despite the phylogenetic proximity to the β-1,3-glucanase from olive pollen (Ole e 9), there is only a 39% similar identity between the two sequences [81]. This is just one of the many studies conducted to investigate the homology between allergens. In addition, another constant in the studies is the discovery of new allergens in pollen.

The existence of various birch reports and cross-reactivity studies has allowed reviews to be written on Bet v 1 and Bet v 2. Because the extensive cross-reactivity and pollen season increases the time of illness in allergic rhinitis and asthma from exposure to sequential birch-related allergens, the sources of allergens should be classified into antigenically related “homologous” groups based (as we have seen) on the molecular properties of both proteins and comparable carbohydrates. Allergen sources should be classified into antigenically related “homologous” groups based (as we have seen) on the molecular properties of both proteins and carbohydrates that are comparable to both the birch homologous group and many other homologous allergens [82].

There are also studies that found that ash and its Fra e allergens share homology with several other plant species [83]. There are also reports of the homology and cross-reactivity of fruit, vegetable, and tree allergens with the three groups of homologous allergens of Bet v 1, profilins, and lipid transport proteins (LTPs), and to plant glycoprotein hydrocarbons that are frequently found to be IgE-directed against these antigenic determinants. The homologous allergens of Bet v 1 in apple (Mal d 1) have a high homology, with 65% identity in the amino acid sequence and 56% identity at the nucleic acid level. In addition to cross-reacting, Mal d 1 shares IgE epitopes for hazelnut (Cor a 1), pear (Pir c 1), apricot, cherry (Pruav 1), plum, celery (Api g 1), carrot (Dau c 1), parsley, and potato [84].

On the other hand, there are also recent reports of the identification of allergenic proteins using these techniques. In the publication of Huerta Ocampo J. A and LM Teran in 2020, they note that they detected, purified, and characterized eight proteins (including Enolase 1, chloroplastic Enolase 1, xylose isomerase, and others), and identified them with a proteomic methodology. That is, 2-DE—two-dimensional electrophoresis; isoelectric focusing; IPG—immobilized pH gradient; immunoproteomics; HPLc—high-performance liquid chromatography; MS—mass spectrometry; and MS/MS Tandem [85]. Alternatively, the identification of allergenic proteins of pecan nut pollen (Carya illinoinensis; CaI i 1, CIr i 2 and Iar i 4) has also been carried using the same proteomic techniques [86], as has the identification and characterization of a new Pecan Car i 2 [87]. Furthermore, the identification of new short ragweed pollen allergens—Amb a 1 and Amb a 3—as well as seven new candidate allergens has been realized using combined transcriptomic and immunoproteomic approaches [88]. One study reports the identification, characterization and sequencing of six new ash pollen allergens (Fra e 2, Fra e 3, Fra e 6, Fra e 10, Fra e 11, and Fra e 12) using immunoproteomics [89].

Nuñez Borque E. and colleagues selected patients with positive skin tests and/or specific IgEs who were allergic to banana pollen, to obtain their sera for evaluation and conduct tests for the confirmation, elicitation, and evaluation of the reaction. The sequence of tests were as follows; Skin Prick Test (commercial series: ALK-Abelló SA); ImmunoCAP System FEIA to measure IgE; ELISA for the purification of Pla a 1 and Pla a 2; SDS-PAGE, immunoblot analysis; protein separation; and the identification and characterization of Pla a 1 and Pla a 2 and other proteins using mass spectrometry (MS) based on liquid chromatography MS in tandem (LC-MS/MS). They obtained nine IgE-capturing bands in the shade plantain pollen extracts. A 45 kDa band corresponded to Pla a 2; 18 kDa (Pla a 1) and 9 kDa (Pla a 3) bands were recognized in 44.7% and 27.3% samples, respectively; and the 27 kDa protein, a glutathione S-transferase, was identified [90]. This workflow is standard for allergen identification in almost all publications.

### 3.6. Other Methods

One of the most commonly used methods with mass spectrometry is high-resolution two-dimensional electrophoresis, which separates proteins from different samples and identifies differentially expressed proteins when combined with staining and mass spectrometry. Additionally, the stable isotope labeling of two different complex mixtures for proteins is commonly used. The method consists of labeling proteins with isotopes and digesting them to produce labelled peptides [91]. Additionally, hybrid technologies use the antibody-based purification of individual analytes and then mass spectrometry analysis for identification and quantification. Techniques include mass spectrometric immunoassay (MSIA) and the standard capture of stable isotopes with antipeptide antibodies (SISCAPA). Microarrays are also used to study proteins on a large scale, such as with the matrix metalloproteinase (MMP) MMP assay kit [92]. The conformational dynamics of proteins can also be studied by means of hydrogen/deuterium exchange-mass spectrometry (HDX-MS), mainly in the loop-helix portions [93].

## 4. Informatics for the Development of Protein Models

Proteomics data are used to decipher and algorithmically predict proteins and to find associations between an existing protein and a modified protein, knowing its functionality and, if necessary, changing it. They can also be used for the marker discovery process, designing pharmacotherapies in immunotherapy, predicting de novo protein functions, and modeling potential proteins to be used.

**Sequent** is a program that individually analyzes, identifies, and validates proteins of the mass spectrum in tandem. This program evaluates and calculates the sequences of the peptides present with respect to a database. The identification of pollen peptides by the Sequent program is scarce or nonexistent; however, the usefulness of these resources should be further promoted. [94]. Proteomics involves the identification of proteins, peptides and amino acids through MS/MS sequences. **Comet** is a search engine for databases and corresponds to an open-source tool that is freely available [95]. **Mascot** Server is a powerful free service tool that runs on the website and is ideal for identifying proteins and peptides from primary sequence databases in informatics program or using mass spectrometry data. This tool allows for the visualization of chemical and post-translational modifications in any direction (up, down, lateral), which can be quantified by means of isobaric labeling, spectral libraries, cross-linking, and intact peptides [96,97].

### 4.1. Protein Data Bank

For the design, consultation, and visualization of a protein, RCSB PDB (RCSB.org) consists of a global Protein Data Bank (PDB) and contains 3D structure data for the design of large biological molecules (proteins, DNA, and RNA). It is one of the first open-access digital resources widely used in molecular biology and the medical fields. Its capacity allows for the study of allergies by consulting 3D structure data for proteins in the environment, urine, and desquamation of mites, pollen, food, and other elements. This database allows users to find any protein structure in any organism on the planet. The forms of IgE and its probable epitopes can be determined after sequencing the target protein [98].

Some publications report the usefulness of the Protein Data Bank program in the case of tree allergen epitopes, the subject of the mountain cedar (Juniperus ashei) Jun a 3, a pollen epitope designed by the template to build homology models for the allergen. The programs were used to extract the distance and dihedral angle constraints from the protein database files and determine the minimized energy structures. This Jun a 3 model has features common with those of completely different protein families, which has raised the suspicion that other structural characteristics may mediate the response, as we already saw in the case of allergen carbohydrates [99].

This tool has become essential for research and education in medicine, biology, biotechnology, and other related science fields since access to the large quantity of 3D structure data stored in the PDB contributes significantly to the progress of the study of almost any structure [100].

### 4.2. Immune Epitope Database (IEDB)

The Immune Epitope Database (IEDB), in conjunction with the Protein Data Bank, has been used to identify epitopes. It is a publicly available NIH-NIAID-funded database of T and B cell epitopes selected from the published literature or from the direct submission of large-scale NIH-NIAID-funded epitope discovery contracts. From the home page, epitopes can be searched for using selected criteria. However, we found no pollen papers using this resource. The interest in studying or predicting the TCR for the corresponding epitope and having particular HLA/epitope combinations should be a very promising area, but at this time is not yet ready for the implementation of epitope identification of this type because of the sparse literature [101].

### 4.3. UniProt

UniProt is a high-quality tool used to access protein sequences; conduct identification mapping, sequence alignment, and peptide searches; predict a sequence obtained from previous studies; and find homology between sequences and proteins with defined sequences. More than 190 million sequences in UniProtKB exist today. The database reduces redundant searches, which provides greater specificity and reliability for researchers [102].

In a study by Stratilová, B., et al., the UniProt database indicated a sequence identity of 54.6–60.5%, suggesting that this protein from *Petroselinum crispum* is a novel allergen. It was structurally characterized and named Pet c 1, a PR-10 defense protein [103].

### 4.4. Swiss-Prot

In a collaboration between the Swiss Institute of Bioinformatics, the European Bioinformatics Institute (EBI), and the Protein Information Resource (PIR) in 2002, the UniProt consortium was created. Swiss-Prot, together with TrEMBL, both automatic resources, also teamed up with the PIR to produce the world’s most important protein catalogue: UniProt Knowledgebase. In addition, the Swiss Institute of Bioinformatics has its website Swiss-Prot, which is a biological database of protein sequences and a fully automated server for modeling the homology of protein structures [104]. ExPASy is also found here, which can analyze protein sequences and structures and 2D-PAGE. ExPASy has two important tools: “pIcarver”, which is a tool to visualize theoretical distributions of peptides by isoelectric points in a certain pH range and proposes a fractionation scheme that generates fractions with similar peptide frequencies, and MALDIPepQuant, a tool to quantify the MALDI peptides (SILAC) from Phenyx [105]. Moreover, Expasy has the Swiss Bioinformatics Resource Portal, where users can find information for various techniques including genetics, metagenomics. transcriptomes, and cell cultures.

The cross-reactivity of respiratory allergies in the clinic can be observed with both inhaled allergens and food allergens in oral allergy syndrome (OAS). For this reason, Michael Platt, et al. performed a high-throughput analysis with all known allergy epitopes within the Immune Epitope Database (IEDB; http://www.iedb.org, accessed on 21 June 2023) for five common species of five subclasses of inhalant allergens and compared their sequences with those of food allergens. In a subsequent step, foods with a known cross-reactivity were searched in the European Molecular Biology Laboratory-European Bioinformatics Institute (EMBL-EBI) protein database (http://www.uniprot.org, accessed on 21 June 2023) with programs that allowed exact matches.

The results showed that 23 common inhalant allergens had 4429 unique epitopes; the 19 foods implicated in ODS had 4497 protein sequences. The algorithm used was con(BLAST), which identified cross-class and intra-class sequence similarities for the five inhalant allergy classes, with a high similarity for mites, grasses, and trees and cross-reactive inhalant allergy epitopes. The method proved to be very efficient [106].

### 4.5. Modeler

Modeler is a informatics program that looks for the homology or comparative modeling of three-dimensional structures between proteins, which when compared, predicts the three-dimensional structure of a given sequence. This application automatically calculates a guide that has all the atoms except hydrogen. It implements the comparative modeling of protein structures by satisfying spatial restrictions according to the user. It is based mainly on its alignment with one or more proteins of the known structure, and it is compared with computer templates. It can also perform the de novo modeling of loops in proteins, the alignment of multiple sequences and/or protein structures, the optimization of protein models with respect to a function, and other tasks [107].

### 4.6. Rosetta Commons

The Rosetta Commons application contains algorithms for the modeling of protein structures and the comparison and analysis of structures. One of the characteristics of the program is the computational coupling that explains the inherent flexibility of protein monomers to simulate the conformational selection of pre-generated sets. In other words, if an amino acid moves chirally, the program detects this modification and couples it. This informatics program has contributed to remarkable advances in biology and medicine. There are reports of de novo protein design, prediction of biological macromolecule structures, enzyme design, ligand coupling, and macromolecular complexes [108].

### 4.7. AlphaFold Protein Structure Database

The AlphaFold Protein Structure Database from the EMBL-EBI is an open-access program that provides more than 200 million highly accurate protein-structure predictions. Its database has expanded the programmatic and visual interaction of protein structures through atomic coordinates that can be predicted with very high prediction and confidence intervals. 

It creates residual and paired models and predicts alignment errors with precision—this has never been seen before in the Critical Evaluation of Protein Structure Prediction (CASP). It has been used in the fields of bioinformatics, structural biology and drug discovery and will undoubtedly widely serve the study of allergic diseases [109,110].

It can be used to identify allergens in tree pollen, extract proteins from pollen and perform mass spectrometry, and then perform Swiss-Prot identification of similar proteins with another protein; and identify query sequences with discontinuous, similar sequences, or on a peptide basis. It is still not well defined in some cases of respiratory allergens whether the homology is in the protein or in the carbohydrates of the allergen, or whether there is another structure that encourages cross-reactivity. This is because of molecules that are not homologous, such as food allergies in individuals that develop respiratory symptoms. The utility of this tool, Swiss-Prot, or AlphaFold Protein Structure Database, cannot help us to define whether it is the sequence of the molecule or the structure.

### 4.8. X-ray Cryptography

Another among the older technologies that continue to be used and that were the pioneers of proteomics, X-ray cryptography stands out. It provides not only the three-dimensional structure but also a deeper understanding of protein structure and function. Almost 86% of the data in the common databases have been generated by this technique. In general, there are two methods for this technique: (1) Vapor diffusion, which consists mainly of the protein and a precipitating mixture that is dehydrated to concentrate the protein crystal with balanced osmolarity in a nucleus, before radiological study is applied. (2) The batch method consists of bringing the protein directly to the nucleation zone and combining it with a precipitant until it is crystallized with oil [111]. The interest of this is looking for proteins that are involved in chemotaxis and the inhibition of the immune response by the cells involved in the allergic reaction. Several cell types must migrate from the peripheral blood to the zone of stimulation or to the zone of extravasation during contact with the allergen, which is represented by a rash. The inhibition or blocking of cell migration is an approach that this group has for the control of the type-2 inflammatory response, type-1 hypersensitivity, the Th2 response, and anaphylaxis.

## 5. Discussion

Aspects such as homology between allergens, cross-reactivity, and the discovery of new allergens through proteomics have been reviewed. The combination of reports with immunotherapy or therapy remains, in addition to the objectives already mentioned. A priority, and from my point of view a fundamental perspective, is to discover whether it is a treatment to alleviate symptoms (conventional pharmacology) or immunotherapy, which has the possibility to cure the patient. The vision of acting on the developmental pathways of hypersensitivity is not new; however, determining a means to represent one or several targets in this type-1 hypersensitivity pathway may provide an idea for action in terms of providing other therapy alternatives.

Proteins in respiratory allergy diseases are of paramount importance, and we believe that the allergic response can be controlled at three points, which are called the immune lockout systems. The first step consists of identifying the specific paratope and epitope of the given allergy problem with proteomic methods, with the intention of blocking or occupying it, either with an antagonist or an antibody. This process is already happening, as we mentioned above, with the antibodies benralizumab, dupilumab, mepolizumab, omalizumabm and reslizumab, which are used in asthma. Ther is also another blocking system that consists of preventing the cell migration of mast cells, basophilsm, and eosinophils to the site of attachment to the epithelium or endothelium [112].

However, although models exist for the study of migration and for the blockade of some molecules, the blockade of some sites, such as those suggested above, has not yet been studied. The aim of this study was to explore which fields of respiratory allergic diseases use proteomics, and to reveal that several proteins involved in the different allergy processes, the homology of allergenic proteins, cross-reactivity, and epitope determination, are the most studied lines, alongside the already-described tree allergens.

## Figures and Tables

**Figure 1 ijms-24-12924-f001:**
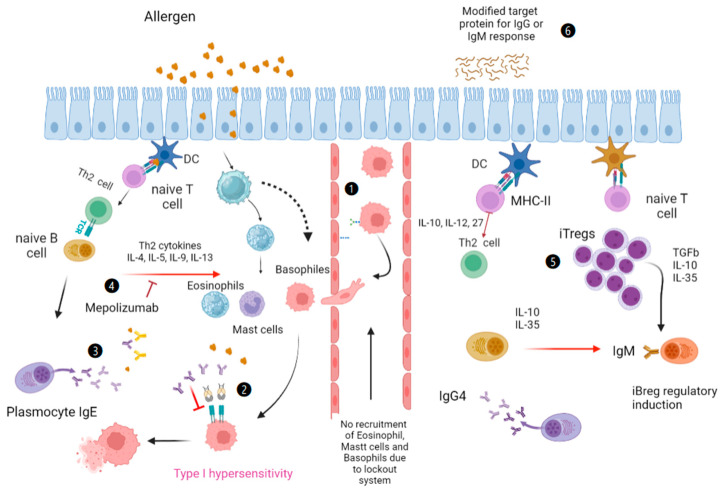
Lockout system. (1) Migration blockade, (2) FcεRI Lockout. (3) Specific antigen-epitope or paratope-specific blockade. (4) anti-interleukin antibody blockade. (5) Recombinant cytokines. (6) Stimulation and monitoring of different proteins. Created with BioRender.com (accessed on 12 Jun 2023).

## Data Availability

Not applicable.

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
