# Peer review of "Perspectives of Proteomics in Respiratory Allergic Diseases"

_ijms, 2023, doi:10.3390/ijms241612924_

Round 1

Reviewer 1 Report

When I first read the title and abstract, I though this review paper will talk a lot about current studies of “Respiratory Allergic Diseases” using Proteomics methods. However, I am disappointed after reading the whole paper, I didn’t get the information about the current issue in Respiratory Allergic Diseases, and how the proteomics helped to solve the issues. Here are some main problems I found about this review paper.

1, most of writing does not related to “Respiratory Allergic Diseases” except the section 2 (Global characteristics of allergies for proteomics studies) and section 5 (discussion). While the rest of parts are just overall talk about the methods used in proteomics, if we don’t look at the section 2 and 5, we can call this review paper “The concepts of proteomics”. Another main problem is that for each method, the author just explained what it is, and didn’t talk about how this method is used in Respiratory Allergic Diseases related studies. I think the author should add some examples about previous studies using each method, and maybe also talk about the pro and con of the method.

2, Another issue about section 3 (Proteomic technologies for the study of respiratory allergies) and 4 (Informatics for the development of protein models), after my reading, I feel that the author just throws out all kinds of protein related or biochemistry related techniques on the paper, without a logical order, or proteomics workflow. I suggest the author should create a story line, for example, the author talk a lot of the pollen proteins, the SDS or western blot can be first used to identify the proteins, then if next question is where is the epitopes, point mutation, LC-MS/MS can be used to detect, informatics like Mascot, PDB, Resotta/Alpha fold can be used to model the binding structure. The author should clear what kind of problems that those methods can answer, the problems should be “Respiratory Allergic Diseases” related.

3, The author used the whole section 1, introduction part, to talk about the history and background of “proteomics”, again, this only partially reflects the title “Perspectives of Proteomics in Respiratory Allergic Diseases”. I think the author should talk about the overall application of proteomics in studies about Respiratory Allergic Diseases”. Then seamlessly connected to section 2 (Global characteristics of allergies for proteomics studies). The author needs to reorganize the writing about this part too. For example, the author first talk about Tregs, nTregs, iTregs and so on in the fourth paragraph on section 2, then talk about Immunoregulatory responses and then bring out those cells again. I feel like the author can delete the forth paragraph ( line 4 -14 on page 3). In the fifth part, discussion part, the second paragraph, again, repeatly talk Tregs, nTrges and so on, I feel this is no necessary. In the discussion part, the author should bring out more “unsolved” “urgent” subjects or issues in Respiratory Allergic Diseases that may be potential solved using proteomics method.

4, Another issue I noticed that for most of paragraphs, the author only has one reference paper at the end of that paragraph. As a review paper, it should cover very broad aspects of the topics in each paragraph, and I believe there are tons of papers related to each subject, the author should try to add more most representative references.

5, Some minor writing issues: keyword part on 1st page, double colon after “Mascot” .  Section 3.3. Western blotting, the first sentence “Is one of the most ….”

English is good and easy to understand. 

Author Response

Response to Reviewer 1 Comments

Point 1: most of writing does not related to “Respiratory Allergic Diseases” except the section 2 (Global characteristics of allergies for proteomics studies) and section 5 (discussion). While the rest of parts are just overall talk about the methods used in proteomics, if we don’t look at the section 2 and 5, we can call this review paper “The concepts of proteomics”. Another main problem is that for each method, the author just explained what it is, and didn’t talk about how this method is used in Respiratory Allergic Diseases related studies. I think the author should add some examples about previous studies using each method, and maybe also talk about the pro and con of the method.

Response 1: I welcome your comments and guidance. The requested concepts have been added (in red) throughout the manuscript, where the usefulness of proteomics as a tool to discover respiratory allergens, mainly by trees and animals, has been discussed. Homology between allergens from different families and cross-reactivity. As one of the main approaches to respiratory allergies, and our idea of the perspective to treatment, in the end, different from the currently used immunotherapy.

Point 2: Another issue about section 3 (Proteomic technologies for the study of respiratory allergies) and 4 (Informatics for the development of protein models), after my reading, I feel that the author just throws out all kinds of protein related or biochemistry related techniques on the paper, without a logical order, or proteomics workflow. I suggest the author should create a story line, for example, the author talk a lot of the pollen proteins, the SDS or western blot can be first used to identify the proteins, then if next question is where is the epitopes, point mutation, LC-MS/MS can be used to detect, informatics like Mascot, PDB, Resotta/Alpha fold can be used to model the binding structure. The author should clear what kind of problems that those methods can answer, the problems should be “Respiratory Allergic Diseases” related.

Response 2: We have introduced the necessary changes in some sections of No. 4. We have introduced quotes from articles with descriptions of what they did with the help of the programs.

At the end of AlphaFold Protein Structure Database, which would be done to identify allergens in tree pollen, and to define whether allergen homology is the cause of cross-reactivity.

Point 3: The author used the whole section 1, introduction part, to talk about the history and background of “proteomics”, again, this only partially reflects the title “Perspectives of Proteomics in Respiratory Allergic Diseases”. I think the author should talk about the overall application of proteomics in studies about Respiratory Allergic Diseases”. Then seamlessly connected to section 2 (Global characteristics of allergies for proteomics studies). The author needs to reorganize the writing about this part too. For example, the author first talk about Tregs, nTregs, iTregs and so on in the fourth paragraph on section 2, then talk about Immunoregulatory responses and then bring out those cells again. I feel like the author can delete the forth paragraph ( line 4 -14 on page 3). In the fifth part, discussion part, the second paragraph, again, repeatly talk Tregs, nTrges and so on, I feel this is no necessary. In the discussion part, the author should bring out more “unsolved” “urgent” subjects or issues in Respiratory Allergic Diseases that may be potential solved using proteomics method.

Response 3: Thank you for your comments.

Part of the old introduction (in blue) has been removed,

We spoke at the end of the introduction according to your comment, to talk about respiratory allergic diseases, to connect with section 2.

Removed line 4-14 (in blue).

Substantial changes were made to the discussion (deleted in blue, added in red).

Point 4: Another issue I noticed that for most of paragraphs, the author only has one reference paper at the end of that paragraph. As a review paper, it should cover very broad aspects of the topics in each paragraph, and I believe there are tons of papers related to each subject, the author should try to add more most representative references.

Response 4: The problem of quotations has been solved and almost in every topic they have been introduced.

Point 5: Some minor writing issues: keyword part on 1st page, double colon after “Mascot” Section 3.3. Western blotting, the first sentence “Is one of the most ….”

Response 5: The problem of Mascot scoring has been solved. And it was changed in Western blotting by; The most used techniques for separation.

Reviewer 2 Report

The paper does not add much to the field. I mean, it does not show anything that would not be known, does not show any new view, synthesis, analysis. And the future directions selected by the author are a kind arbitrary. 

Besides, it does not have a content. It is based on 51 references only and has only a single figure.

Minor improvements required. 

Author Response

Response to Reviewer 2 Comments

Point 1:

Response 1: I appreciate your comments.

The manuscript has been substantially modified.

Concepts have been added (in red) throughout the manuscript, where the usefulness of proteomics as a tool for discovering respiratory allergens, mainly from trees and animals, has been discussed. Homology between allergens from different families and cross-reactivity. As one of the main approaches to respiratory allergies, and our idea of the perspective to treatment, in the end, different from the currently used immunotherapy.

We have introduced the necessary changes in some sections

We have introduced quotes from articles with descriptions of what they did with the help of the programs.

At the end in Protein Data Bank and AlphaFold Protein Structure Database,

We added concepts such as; What would be done to identify allergens in tree pollen, and to define if allergen homology is the cause of cross-reactivity.

Part of the old introduction (in blue) has been removed and we spoke at the end of the introduction of respiratory allergic diseases, to connect to section 2.

Line 4-14 (in blue) has been deleted.

Substantial changes have been made to the discussion (deleted in blue, added in red).

The problem of citations has been solved and they have been introduced in almost all topics.

A table has been added.

Round 2

Reviewer 1 Report

Since this a revised version, I only focused on the the edited part by the author. This time, the author added more up-to-date and closely related work/reference, this is good. However, there are still a lot of minor issues and the author should fix them, especially about how to write more clearly and to not misleading. Here are some examples: 

1, Line 77, double space before "In allergic ...."

2, Missing reference number 7, 8, 9, 13. (the author should double check all the numbers) Line 322, there should not be a full stop before reference 40

3, Line 215, “(MC)” after mast cells can be deleted, since in previous text, the author only used “mast cells”

4, Line 343-349, this paragraph is more like a background for IgE, doesn’t fit here. The author may consider move this to the introduction part, somewhere before mentioning IgE the first time.

5, Line 350-353, this seems to be Spanish, not English, please correct the language. Line 354-357, this seems to be same as Line 350-362, repeat. Line 358, after “main allergen”, consider add “for example” or “namely” to help make it clear. Line 360-262, “others that have been identified….” it is not clear what are they

6, Some other sentences that is misleading or can be fixed

Line 392, “since due to”

Line 397, “many data ..have been”

Line 403, “Valenta, R et al. explore three of the best-known groups are”

Line 409-410, “Toshihiro Osada, 409 Mitsuhiro Okano so report them …”

Line 416, “It has been seen that … but also…” this sentence is very misleading

Line 447 “they report in their analysis the characterization of 8 proteins..”

7, Line 452-458, those are not completed sentences.

8, Line 459 “Nuñez Borque E. and cols”, do you mean “Nuñez Borque E. and coworkers”? Line 461-463, the author should be more clear about each step, it is hard to understand each step based on current writing.

9, Line 529, the punctuation mark before “however”

10, Line 540-541, “a sequence identity 540 of 54.6-60.5%, suggesting this protein…” this is not clear which protein the author is talking about, and use which protein for sequence identity?

Line 558-570, can be shorten to one paragraph, Line 557,  full stop before “transcriptome”?

Line 601-603, those are not completed sentences if the author try to read it through. 

The writing need to be improved significant. There are a lot of parts in the paper is not even a full sentence. Also, the author should try to use professional transition words when talk about other people's work, instead of just throwing out a work by other researcher. Try to make paragraph to paragraph connecting more smoothly. for example, in a lot of part of this paper, the author talks about a protein or a topic, it is hard to tell what the author is talking about, unless going back to previous text. 

Author Response

Response to Reviewer 1 Comments

Since this a revised version, I only focused on the the edited part by the author. This time, the author added more up-to-date and closely related work/reference, this is good. However, there are still a lot of minor issues and the author should fix them, especially about how to write more clearly and to not misleading. Here are some examples:

Response : Everything that needed to be deleted has been removed and the changes can now be viewed with the change control.

Minor details or problems have been revised, resolved and some parts of the manuscript have been written more clearly.

Point 1: Line 77, double space before "In allergic ...."

Response 1: Space has been eliminated page 2, line 54.

Point 2: Missing reference number 7, 8, 9, 13. (the author should double check all the numbers) Line 322, there should not be a full stop before reference 40.

Response 3: The references have been checked and corrected with the missing reference: number 7 (line 57), 8 (line 63), 9 (line 72), 13 (line 104).

Deleted the dot before reference 40 now 44 (line 293).

Point 3: Line 215, “(MC)” after mast cells can be deleted, since in previous text, the author only used “mast cells”.

Response 3: Line 215, deleted "(MC)" after mast cells, now line 179.

Point 4: Line 343-349, this paragraph is more like a background for IgE, doesn’t fit here. The author may consider move this to the introduction part, somewhere before mentioning IgE the first time.

Response 4: The paragraph that was on line 343-349 was moved to the place where the IgE narrative first appears, now line 191-197.

Point 5: Line 350-353, this seems to be Spanish, not English, please correct the language. Line 354-357, this seems to be same as Line 350-362, repeat. Line 358, after “main allergen”, consider add “for example” or “namely” to help make it clear. Line 360-262, “others that have been identified….” it is not clear what are they

Response 5: It was rewritten from line 352-358. These modifications remained on line 312-318.

Included on line 358 "namely", currently on line 319.

Also on lines 360-363 included; and allergens have also been reported; Mus m 1, Mus m 2, from mice also identified by liquid chromatography. Currently on line 320-321.

Point 6: Some other sentences that is misleading or can be fixed.

Response 6:

Line 392, “since due to”

Replaced by: Most studies have focused on identification, homology between families and allergenic sequences, as there is cross-reactivity between allergenic proteins due to their similarity. 357-359 Now.

Line 397, “many data ..have been”

Response Replaced by; “There is a lot of data on homologies”, 363 Now.

Line 403, “Valenta, R et al. explore three of the best-known groups are”

Response: This paragraph has been reworded, and now corresponds to the line 369-374

Line 409-410, “Toshihiro Osada, 409 Mitsuhiro Okano so report them …”

Response: The paragraph has been restructured, now it is line 377-382.

Line 416, “It has been seen that … but also…” this sentence is very misleading

Response: The paragraph has been restructured, it is now on the line 387-393.

Line 447 “they report in their analysis the characterization of 8 proteins..”

Response: The sentence has been changed, it is now the line 417-423.

Reviewer 2 Report

Indeed, it is a different work now, potentially having sufficient content and some ideas behind.

Comments (no special order):

1.      Explain “FcεRI” upon first appearance, separately in the abstract and the main text. Unify “FcεR1” to “FcεRI”. The same for all abbreviations.

2.      Avoid typos, e.g. “com-plex” in line 216.

3.      Suggested recent papers to be included:

a.      PMID: 34067047 while discussing IL-10, FcεRI and MC.

b.      PMID: 33926084 while mentioning biologicals.

c.      PMID: 35628549 while discussing the importance of Tregs, especially iTregs, in allergies.

d.      PMID: 36674705 while discussing allergens and related diagnostics.

e.      PMID: 33805442 while discussing fruit allergens.

4.      References 7-9 are missing in the text?

5.      Figure 1 must have better quality, i.e. resolution.

Minor amendments

Author Response

Response to Reviewer 2 Comments

Point 1: Explain “FcεRI” upon first appearance, separately in the abstract and the main text. Unify “FcεR1” to “FcεRI”. The same for all abbreviations..

Response 1: Response 1: Thank you for your comments. You will be able to see all the changes in change control.

  1. In the summary, added FcεRI is a high affinity receptor (page 1 line 13).

The first time in the text it was noted:

"FcεRI this receptor is a member of the Ig superfamily, an antibody isotype involved in sig-nal transduction." (page: 4 line: 174, 175)

Unified in all the text to: FcεRI

Point 2: Avoid typos, e.g. “com-plex” in line 216.

Response 2: These spelling errors were avoided and the existing ones were repaired.

Point 3: Suggested recent papers to be included:

Response 3: were included

  1. PMID: 34067047 while discussing IL-10, FcεRI and MC. References: [24].
  2. PMID: 33926084 while mentioning biologicals. References: [9].
  3. PMID: 35628549 while discussing the importance of Tregs, especially iTregs, in allergies. References: [15].
  4. PMID: 36674705 while discussing allergens and related diagnostics. References: [61].

Point 4: References 7-9 are missing in the text?

Response 4: All the references have been reviewed in order and according to their citation.

Point 5: Figure 1 must have better quality, i.e. resolution.R

Response 5: The image quality, resolution and size have been improved. TIF, 1047 X 738, 300ppp.

Round 3

Reviewer 1 Report

The manuscript is significantly improved, agreed to accept in present form. 

Reviewer 2 Report

No additional comments.